# SimpliGuard: Robust Mesh Simplification In the Wild

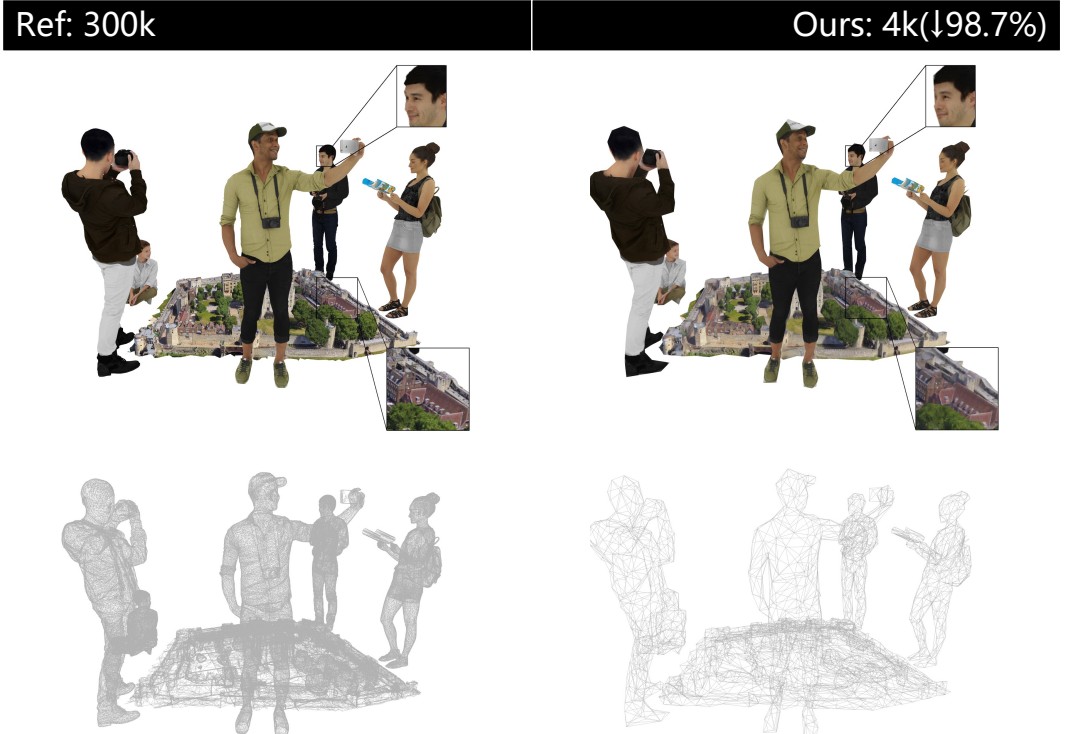

**Figure 1: We propose a framework that enables extreme mesh simplification for arbitrary complex meshes in-the-wild.**

## ABSTRACT

Polygonal meshes are widely used to represent complex geometries. However, the increasing complexity of models often leads to large meshes with millions of triangles, raising significant challenges for storage, transmission, and computation. Mesh simplification, a process of reducing the number of triangles in a mesh while preserving its overall shape and important features, has emerged as an indispensable technique to address these challenges. In this work, we focus on the problem of obtaining a visually consistent ultra-low-polygon mesh for complex meshes. Unlike previous methods, we design a robust simplification framework, SimpliGuard, to handle any meshes in the wild. Firstly, a reconstruction module is used to construct a low-polygon mesh with a similar shape but a manifold topology. Then, a texture initialization module is employed to quickly initialize the entire texture map. After that, a differentiable rendering module is utilized to optimize the overall structure and texture details, ensuring high-quality results. For meshes with skeletons, the correctness of motion can be preserved with our designed motion post-processing module. Experimental results demonstrate that SimpliGuard significantly outperforms previous methods and various featured software, including Blender and Simplygon.

## CCS CONCEPTS

• **Computing methodologies**;

## KEYWORDS

Mesh Simplification,Level of detail,Differentiable Rendering

## 1 INTRODUCTION

Meshes have been widely employed in various domains, such as virtual reality and game development. However, as the complexity of meshes increases, the required computational and rendering resources also increase, thereby limiting their performance on mobile devices, VR, and other edge devices. To address this issue, mesh simplification techniques have been proposed.

Traditional methods, such as QEM (Quadric Error Metrics[Garland and Heckbert 1997]), employ a greedy strategy to perform edge

collapses and merges on the original mesh. To preserve texture correctness as much as possible, some variants [Garland and Heckbert 1998] incorporate the attributes as constraints into the optimization process. However, these traditional methods essentially generate a sub-mesh from the original mesh, regardless of whether the original mesh's structural design is friendly to decimation. This leads to the following issues: 1) the lower bound of the reduced face count is constrained by the structural design of the original mesh; 2) the output quality varies for different meshes, and severe structural errors, such as holes, can occur for meshes in the wild; 3) the texture distortion and fragmentation after simplification result in poor visual perception. Consequently, traditional mesh simplification algorithms are difficult to be directly used, and are often used as auxiliary tools for designers.

Recently, some works based on differentiable rendering, like Hasselgren et al. [2021], have emerged. They utilize differentiable rendering to optimize the texture and topological structure simultaneously. However, these methods only rely on differentiable rendering to optimize the structure and texture, which can lead to the following issues: 1) careful parameter tuning is necessary for different meshes; otherwise, the results are unstable and of varying quality; 2) the optimization process is extremely slow; for instance, Nvdiff requires over half an hour to achieve satisfactory results, which is impractical for real-world usage; 3) for meshes with skeletons, the animation of the simplified mesh cannot be guaranteed.

In this work, a robust framework, SimpliGuard is proposed. With SimpliGuard, a mesh of any complex structure can be reduced to a target number of faces within a few minutes while maintaining similar shape and texture. For meshes with skeletal animation, SimpliGuard guarantees the correctness of the output mesh's motion. The overall structure consists of four parts. The reconstruction module can transform any complex mesh in the wild into a manifold mesh with a similar shape. The texture initialization module quickly generates coarse texture maps, which has two benefits: 1) it significantly reduces the learning time of the differentiable rendering module for textures, and 2) for scenes that do not require close-up observation, these texture maps can be directly used. The differentiable rendering module, following the texture initialization module, further optimizes the structure and texture details to improve the overall quality of the generated mesh. Finally, the motion post-processing module ensures that the animated mesh does not suffer from structural problems even when reduced to a small number of faces. The overall contributions are as follows:

1. We propose SimpliGuard, a framework that can generate high-quality simplified meshes for meshes in the wild. For rigged meshes, SimpliGuard ensures correct motion and eliminates structural issues even after simplification. The advantages are achieved through the combined efforts of various modules.

2. The framework can generate reliable results in a few minutes. This is achieved through our meticulous module design and initialization acceleration.

3. We propose multiple loss functions that significantly improve the results of differentiable rendering, enhancing the usability of the generated meshes.

4. Qualitative and quantitative experiments show that our method is superior to the previous academic approaches and featured software in terms of metrics and visual quality.

## 2 RELATED WORK

*Mesh Simplification.* Traditional mesh simplification algorithms [Luebke 2003] can be roughly divided into two categories: local strategies and global strategies. One representative algorithm of global strategies is Vertex Clustering [Low and Tan 1997; Rossignac and Borrel 1993; Valette and Chassery 2004; Valette et al. 2005]. This kind of algorithm sorts each vertex based on a defined cost function to ensure that less important vertices are more likely to be merged. However, these methods can completely change the topology of the input mesh in an unpredictable way, and the resulting structure is not visually similar to the original structure, which is unacceptable. In the local feature-driven approaches, vertex decimation [Schroeder et al. 1992; Soucy and Laurendeau 1996] iteratively selects a vertex, deletes it along with the surrounding faces, and re-tessellates the resulting hole. However these methods are effective for manifold surfaces and not suitable for in-the-wild scenarios. As another kind of local strategy, edge contraction, proposed by [Hoppe et al. 1993], has been the most commonly used simplification operation. Its core idea is to merge individual edges into a single vertex. The selection of edges and the optimal merged vertex depend on the defined cost function. Consequently, several related algorithms have been proposed. Ronfard and Rossignac [1996] measures the distance between points and planes using a quadric matrix. Inspired by Ronfard and Rossignac [1996], Garland and Heckbert [1997] introduced a new error matrix to approximate geometric deviation, known as QEM. Subsequently, many QEM-based variants have been proposed, including Garland and Heckbert [1998], Hoppe [1999] ,Wu et al. [2004], Li and Zhu [2008], Thiery et al. [2013] and Liu et al. [2017]. Recently, there have been some methods based on neural networks. Potamias et al. [2022] uses a graph neural network to simplify a given mesh in one pass. Hasselgren et al. [2021] utilizes differentiable rendering to simultaneously fit the shape and texture of the target mesh. However, these methods essentially results in a subset of the original mesh, and thus are limited by the original mesh structure and are difficult to handle for complex meshes in the wild.

*Surface Remeshing.* Surface remeshing is a widely researched field [Khan et al. 2020]. A typical method is constructing a Centroidal Voronoi tessellation on the surface of the mesh [Du et al. 1999]. Due to slow convergence, it requires a good initial sampling. Alliez et al. [2003] introduces an error diffusion-based algorithm that effectively finds better initial sampling. But this method operates in the global parameter domain, which may not find suitable parametrization. To address this, Surazhsky et al. [2003] performs local modifications while referencing the geometry of the original mesh. Although it is more efficient and accurate, the sampling quality is not guaranteed. Thus, some new algorithms have been proposed, including Schreiner et al. [2006] based on advancing-front paradigm, Fu and Zhou [2008] based on 2D fast Poisson disk sampling algorithm, and Fuhrmann et al. [2010] based on novel resampling strategies. However, these methods are difficult to directly apply to our task. On one hand, they mainly focus on improving the accuracy of the fitted mesh, which often leads to an unacceptable number of faces. On the other hand, many algorithms only consider the mesh structure and neglect the mesh's texture properties, making it challenging to apply to assets in real-world scenarios.

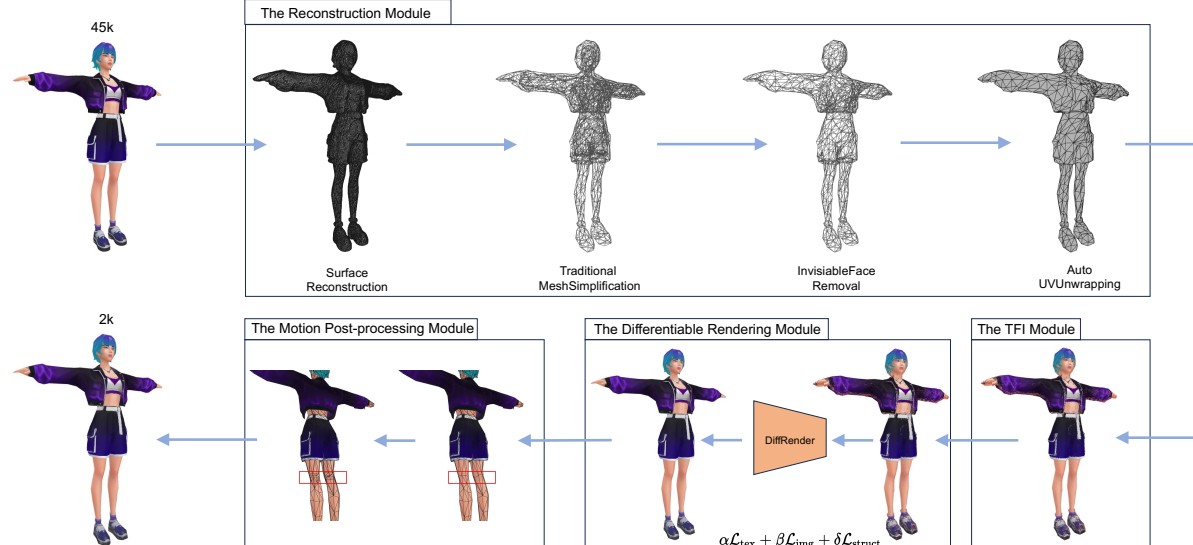

**Figure 2: Overview of SimpliGuard. The framework consists of four modules. Given a high-poly mesh $M_{tar}$, we first apply the reconstruction module to obtain a low-polygon mesh with similar shape and manifold topology. Then, the texture fast initialization module is employed to quickly initialize the entire texture map within few seconds. After that, the differentiable rendering module is utilized to optimize the overall structure and texture details with ours proposed losses. For meshes with skeletons and weights, the correctness of motion can be preserved with the designed motion post-processing module.**

## 3 METHOD

### 3.1 Overview

Given a mesh $M_{tar} = (V_t, F_t, Tex_t, BW_t[opt])$, our goal is to obtain a new mesh $M_{src} = (V_s, F_s, Tex_s, BW_s[opt])$ with a reduced number of faces, where V, F, Tex, and BW respectively represent the vertices, faces, texture maps, and skeletal bone weights. $M_{src}$ is required to be visually similar to $M_{tar}$ as much as possible. Additionally, if there are animated skeletons, $M_{src}$ needs to keep the structural correctness under large deformation. To achieve this objective, we propose a unified framework, SimpliGuard, as shown in Fig. 2.

### 3.2 The Reconstruction Module

Most of the previous QEM-based method achieve decimation by performing edge contraction, essentially resulting in a subset of the original mesh. Take QEM as an example. Without loss of generality, let's assume that during the merging process, $v_1$ is selected as the merged point for $v_1$ and $v_2$. Then, the faces with an edge connecting $v_1$ and $v_2$ will be removed, and $v_1$ will replace $v_2$ in all the faces and edges with $v_2$ as a vertex of them. As a result, the remaining faces are a subset of the original faces with some vertex positions changed. Therefore, for such algorithms to achieve good results on meshes in the wild, the assumption must hold true that a subset of the mesh's faces can well represent the original structure. In reality, however, most meshes do not adhere to this assumption. To address the structural problem, we propose the reconstruction module, which includes Surface Reconstruction, Traditional MeshSimplification, InvisibleFaceRemoval and Auto UVUnwrapping.

*Surface Reconstruction.* Our objective is to obtain a mesh that is similar to the original mesh and has a high-quality structure (watertight and manifold). These properties ensures that the mesh will not suffer from structural issues such as discontinuous triangle faces during optimization. To achieve this goal, we introduce surface reconstruction, which is a well-explored field of research [Alliez et al. 2008; Chen et al. 2023; Khan et al. 2020; Khatamian and Arabnia 2016; Peng et al. 2005]. A typical approach involves transforming the mesh into a signed distance field and then extracting the surface using isosurface extraction techniques, which are subsequently converted into a watertight and manifold triangular mesh [Huang et al. 2018]. This kind of technique generally satisfies our structural requirements. However, considering the requirement for efficient execution, we need to modify these algorithms to minimize memory usage and ensure fast performance. Specifically, when applying the marching cube based methods [Lorensen and Cline 1998], we only retain the relative relationships between the cubes and the mesh. Then, we employ a finer grid representation for the surface regions, while for non-surface regions, a coarser grid representation is used to conserve memory. Additionally, an octree data structure is employed to expedite the computation process by storing information between nodes. As indicated in Figure 7, for a mesh with 30,000+ faces, the reconstruction can be completed within 4.4 seconds.

*Traditional MeshSimplification.* After surface reconstruction, we have obtained a high-quality mesh that is manifold. In this section, our objective is to quickly minimize the face count to achieve the target face count, without strictly pursuing the quality of the reduced mesh. Therefore, we employ the classical QEM algorithm. It should be noted that, since we are reducing the face count of a watertight and manifold mesh obtained through surface reconstruction, various structural issues that may be present in the original mesh are resolved. However, there might be the problem of self-intersection. To address this problem, we additionally propose a curvature constraint. Given $v_i$ and the faces $f_1, \ldots, f_M$ it belongs to,

we compute the normal vectors $N_1, N_2, \ldots, N_M$. Then, the curvature of $v_i$ is calculated as follows: $c_i = 1/\sum_{j=1}^{M} \text{angle}\,(n_i, N_j)$, where $n_i$ is the normal vector of $v_i$ and $\text{angle}\,(n_i, N_j)$ is the angle between $n_i$ and $N_j$. Since the original quadric of each $v_i$ can be written as $Q(v_i) = \sum_{j=1}^{M} \text{area}(f_j) \cdot Q(v_i, f_j)$, where $Q(v_i, f_j) = v_i^T A v_i + 2b^T v_i + d$, we can represent $Q(v_i, f_j)$ using a simple representation [Hoppe 1999]: $Q(v_i, f_j) = (A, b, c)$. Now we need to incorporate curvature into $Q(v_i, f_j)$. Here we directly give the expression, and the proof can be found in the supplementary material.

$$Q_c(v_i, f_i) = \left( \left( \begin{array}{c|c} gg^T & \begin{matrix} \ddots & 0 & \ddots & -g & \ddots & 0 & \ddots \end{matrix} \\ \hline \begin{matrix} \ddots & 0 & \ddots \\ -g^T \\ \ddots & 0 & \ddots \end{matrix} & \begin{matrix} \ddots & 0 & \ddots & 0 & \ddots & 0 & \ddots \\ \cdots & 0 & \cdots & 1 & \cdots & 0 & \cdots \\ \ddots & 0 & \ddots & 0 & \ddots & 0 & \ddots \end{matrix} \end{array} \right), \begin{pmatrix} qg \\ 0 \\ -q \\ 0 \end{pmatrix}, q^2 \right)$$

where $g$ and $q$ can be computed by:

$$\begin{pmatrix} v_1^T & 1 \\ v_2^T & 1 \\ v_3^T & 1 \\ N^T & 0 \end{pmatrix} \begin{pmatrix} g \\ q \end{pmatrix} = \begin{pmatrix} c_1 \\ c_2 \\ c_3 \\ 0 \end{pmatrix}$$

The final $Q_{\text{total}}(v_i, f_i) = Q(v_i, f_i) + \alpha Q_c(v_i, f_i)$, where $\alpha$ is a hyperparameter used to balance QEM and the curvature constraint.

*InvisibleFaceRemoval.* After traditional mesh simplification, there may exist completely invisible faces in the idle state. These faces have no visual impact but increase computational complexity. Furthermore, during animation, these invisible faces can potentially cause self-intersection between the inner and outer surfaces, leading to structural issues. Thus they are necessary to be removed. Here we adopt a straightforward approach to identify the invisible faces. We generate multiple rays from $v_i$ and check for each ray if it intersects with the mesh. If there is no intersection, we increment a counter by 1; otherwise, the counter remains unchanged. Finally, we determine the visibility of the current point by evaluating whether the counter is greater than zero. This process can be expressed as $H_i = \int_{\Omega} \mathbb{I}\,(w)\,dw$. $\Omega$ represents the set of all rays in the hemisphere. $\mathbb{I}\,(w) = 1$ when the ray $w$ doesn't intersect with the mesh, and $\mathbb{I}\,(w) = 0$ otherwise. In practice, we approximate the process by randomly sampling rays from the hemisphere.

*Auto UVUnwrapping.* Until now, the mesh we have obtained lacks UV coordinates, making it unsuitable for texture generation. Therefore, we employ a classic method for automatic UV generation known as Least Squares Conformal Maps [Lévy et al. 2002]. It partitions the surface of the mesh into multiple local patches and minimizes distortion within each patch.

### 3.3 The Texture Fast Initialization Module

We have obtained a simplified mesh with a rough structure that lacks textures. To obtain the texture, an intuitive approach would be to directly optimize the texture using differentiable rendering. However, iterative optimization with differentiable rendering is a time-consuming operation. Considering that $M_{\text{src}}$ and $M_{\text{tar}}$ have similar shapes, we propose a texture fast initialization module based on rendering. In the rendering pipeline, after triangles are rasterized, they are filled with color values sampled from the texture

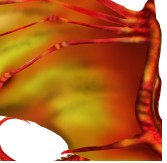 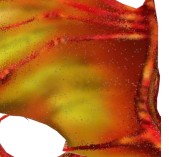 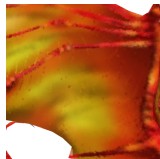

| GT | TFI: Single-Point | TFI: Cycle-Point |

**Figure 3: The Efficiency of TFI: Using the single-point strategy allows for a rough approximation of textures, but often results in numerous "spots". By employing the cycle-point strategy, the occurrence of spots is significantly reduced, leading to an improvement in the quality of the initial texture.**

map based on their UV coordinates. Therefore, each pixel in the rendered image corresponds to a coordinate on the texture map. By simultaneously rendering $M_{\text{src}}$ and $M_{\text{tar}}$ from multiple viewpoints, we can utilize the rendered images of $M_{\text{tar}}$ to fill the texture map of $M_{\text{src}}$. Let $r_t^p \in \mathbb{R}^{h \times w \times 3}$ and $r_s^p \in \mathbb{R}^{h \times w \times 3}$ be the rendered image of $M_{\text{tar}}$ and $M_{\text{src}}$ under viewpoint p. By tracking the relationship between UV coordinates and pixels during the rendering process, we can obtain the pixel-to-uv map $u_t^p \in \mathbb{R}^{h \times w \times 2}$ and $u_s^p \in \mathbb{R}^{h \times w \times 2}$. Then the coarse texture of $M_{\text{src}}$ can be quickly obtained by: $\text{Tex}_s[u_t^p[:,:,0], u_s^p[:,:,1]] = \text{Tex}_t[u_s^p[:,:,0], u_t^p[:,:,1]]$, However, although the obtained coarse texture map can roughly capture the color information of the original mesh, due to the limited rendering viewpoints and resolution, the texture map becomes discontinuous, resulting in numerous "spots" (as shown in Figure 3). To generate a more continuous texture map with the same computational resources, we propose a smoothing algorithm. For each pixel, we select a circular region with a radius of r and centered at the corresponding UV point on $\text{Tex}_t$ and $\text{Tex}_s$. Then, we assign all the pixels within the circular region on $\text{Tex}_t$ to $\text{Tex}_s$:

$$\text{Tex}_s[\text{Cycle}(u_s^p[:,:,0], u_s^p[:,:,1], r)]$$
$$= \text{Tex}_t[\text{Cycle}(u_t^p[:,:,0], u_t^p[:,:,1], r)], \quad \forall 1 \le p \le K$$

As depicted in Figure 3 and Figure 9, this algorithm significantly improves the quality of the initial texture.

### 3.4 The Differentiable Rendering Module

To further improve the quality of the mesh, we introduce differentiable rendering, which contains three types of loss functions: $\mathcal{L}_{\text{tex}}$, $\mathcal{L}_{\text{img}}$ and $\mathcal{L}_{\text{struct}}$. The overall optimization objective is defined as:

$$\underset{\text{Tex}_s, V_s}{\arg\min} \mathbb{E} \left[ \alpha \mathcal{L}_{\text{tex}} + \beta \mathcal{L}_{\text{img}} + \delta \mathcal{L}_{\text{struct}} \right]$$

*Loss of Texture.* To generate a high-quality texture, two requirements should be ensured: 1) The rendered texture colors of $M_{\text{src}}$ should closely match that of $M_{\text{tar}}$. 2) The transitions in the rendered texture should be smooth when observing the mesh closely. Previous research focuses more on optimizing texture color but overlooks the texture smoothness. To address this, we propose $\mathcal{L}_{\text{tex}}$. The loss randomly selects pixel coordinates, called $P_s$, on the texture map during each iteration and applies small random perturbations to these coordinates, resulting in new coordinate points. The objective is to constrain the color of the new coordinate points to be as similar as possible to that of the original points:

$$\mathcal{L}_{\text{tex}} = ||\text{Tex}_s(P_s) - \text{Tex}_s(P_s + \epsilon)||^2, \text{ where } \epsilon \sim \mathcal{N}(0, \sigma^2).$$

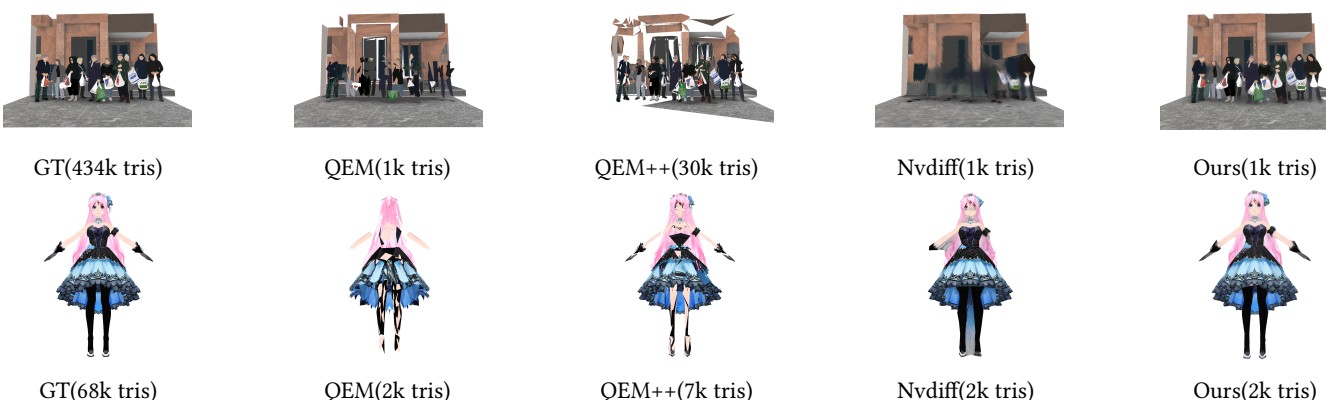

Figure 4: Comparison with academic approaches. QEM and QEM++ suffers from poor quality of the structure and texture. Besides, QEM++ fails to achieve the desired target face number. Nvdiff produces blurry textures and dissimilar structure. In contrast, SimpliGuard achieves best in both structure and texture.

*Loss of Image.* Optimizing texture using MSE loss on rendered images is a common practice in differentiable rendering. However, this approach often struggles to capture fine details in the texture. The reason is that MSE loss does not impose constraints between pixels, which can lead to issues such as unclear edges in the images. To address this problem, we introduce a new loss. This loss calculates the gradients of the rendered image $r_s$ and compares them with the gradients of the target image $r_t$. When the rendered image exhibits blurriness in the edge regions or roughness in smooth regions, it results in larger gradients in those areas. This allows the gradient descent algorithm to effectively optimize the texture in those specific areas. The final loss is defined as follows:

$$\mathcal{L}_{img} = ||\text{Grad}(r_t) - \text{Grad}(r_s)||^2 + ||r_t - r_s||^2.$$

*Loss of Structure.* Optimizing the mesh structure only based on rendering images is unfeasible, as it may lead to issues such as mesh interpenetration, spikes, and surface irregularities. To alleviate this challenge, we impose Laplacian smoothing:

$$\mathcal{L}_{laplacian} = \frac{1}{|V_s|} \sum_{i=1}^{|V_s|} \left\| \frac{1}{|S_i|} \sum_{v_j \in S_i} v_j - v_i \right\|^2$$

Here, $|V_s|$ is the number of vertices in $M_{src}$, and $S_i$ is the neighbourhood of $v_i$. Furthermore, to ensure smoothness in $M_{src}$, we apply a smoothness constraint on the normal vectors:

$$\mathcal{L}_{normal} = \frac{1}{|F_s|} \sum_{(v_1, v_2) \in E_s} (1 - \cos(n_1, n_2))$$

$F_s$ represents the set of faces in $M_{src}$, and $E_s$ represents the set of all connected edges in $M_{src}$. $n_1$ and $n_2$ are the normal vectors of vertices $v_1$ and $v_2$, respectively. $\mathcal{L}_{struct}$ can be represented as:

$$\mathcal{L}_{struct} = w_{normal}\mathcal{L}_{normal} + w_{lap}\mathcal{L}_{laplacian}$$

## 3.5 The Motion Post-processing Module

In real-world applications, many objects are animatable. If these objects cannot be properly animated, they still cannot be used even if the triangles are reduced. Some related research utilizes a neural network to obtain the bone weights [Mosella-Montoro and Ruiz-Hidalgo 2022; Xu et al. 2020]). However, these methods only optimize bone weights or vertex coordinates. It is reasonable for high-poly meshes. But in the case of low-poly meshes, simply moving the coordinates of vertices and optimizing bone weights cannot guarantee the absence of motion artifacts. In this section, we introduce a simple post-processing module. Firstly, given $V_t$, $F_t$, and $BW_t$, we compute $BW_s$ for $V_s$ through the K-nearest neighbor algorithm:

$$BW_s(i) = \sum_{j \in Neighbor_r(i)} \left(1 - \frac{dist(V_s(i) - V_r(j))}{\sum_{k \in Neighbor_r(i)} dist(V_s(i) - V_r(k))}\right) * BW_r(j)$$

Then, we identify the planes where the joints lie and split the faces on the mesh that intersects with these planes. Specifically, we find all points in $V_s$ where the difference in the first two dimensions of $BW_s$ is smaller than $\delta$. Then, based on the physical properties of $BW_s$, these points can be clustered into J groups, and the average of each group's points will serve as the center point for each of the J joints. Based on the orientation of the joints, the normal vectors of the planes can be calculated. With the points and the normal vectors, we can determine J joint planes, which can be used to calculate the intersection points between the planes and the mesh. Finally, these points are used as new vertices to split the faces.

## 4 EXPERIMENTS

### 4.1 Experimental Setup

*Dataset.* To fully validate the effectiveness of the algorithm, we collect 884 assets from Sketchfab and convert them into .obj file format. Details can be seen in the supplementary material.

*Metrics.* We measure the visual quality of the low-poly mesh based on PSNR and SSIM. To better measure perceptual similarity, we design a metric called NLPIPS (Normalized LPIPS), which is based on LPIPS: NLPIPS = $1 - \text{LPIPS}(I_{src}, I_{tar})/\text{LPIPS}(\text{Noise}, I_{tar})$, where Noise is the noise image. All these metrics are computed with the images rendered by 80 camera views. To measure structural similarity, we use the 3D IoU metric [Ravi et al. 2020].

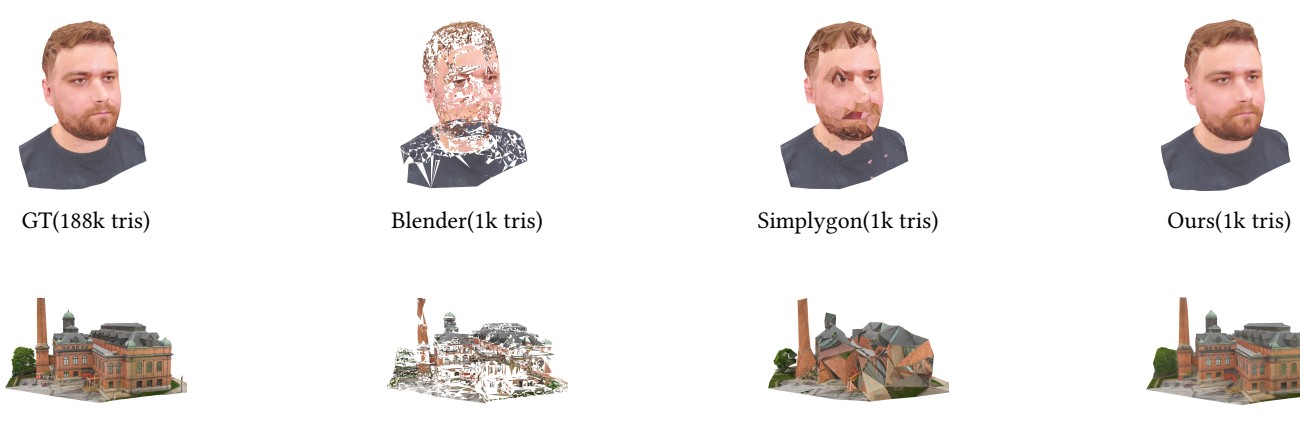

GT(188k tris)      Blender(1k tris)      Simplygon(1k tris)      Ours(1k tris)

GT(997k tris)      Blender(1k tris)      Simplygon(1k tris)      Ours(1k tris)

**Figure 5: Comparison with featured software. Blender fails to handle complex meshes, resulting in fragmented structures in the generated output. Simplygon often leads to texture distortion. In contrast, SimpliGuard consistently produces excellent results even when dealing with complex high-poly meshes.**

## 4.2 Evaluation of the Simplified Meshes

We conduct a comparative analysis with five algorithms, and set three target levels: 1,000 faces, 2,000 faces, and 4,000 faces, which are the typical number of faces that can be rendered in VR scenes for a large number of assets. Since the face count for each asset in the dataset is greater than 30,000, the reduction ratio is at least approximately 86.7%. All results from SimpliGuard are obtained using the same parameters (details in the supplementary material).

*Comparisons with academic approaches.* We select two traditional methods, QEM [Garland and Heckbert 1997] and QEM++ [Garland and Heckbert 1998], and one differentiable rendering-based method, Nvdiff, as comparison benchmarks. Since QEM++ often struggles to achieve the target face count, we calculate the metrics for QEM++ with the meshes that it can simplify to below 5000 faces, which counts for 773 in total. As shown in Table 1, SimpliGuard significantly outperforms other approaches in all metrics. Additionally, it is worth noting that even for other methods evaluated at 2,000 faces, their results do not surpass our method at 1,000 faces.

*Comparisons with industry software.* We select two widely used industrial software, including Blender and Simplygon, as comparison benchmarks. As can be seen in Table 1, our method still achieves the best performance across all metrics.

*Visualization analysis.* To visually compare different methods, we randomly show some results in Figure 4 and Figure 5. It can be observed that QEM and QEM++ exhibit rough structures and distorted textures. Nvdiff, Blender, and Simplygon also fail to achieve similar 3D structures or high-quality textures. In contrast, our method achieves the best results in terms of both structural fidelity and texture clarity. This observation aligns with the quantitative metrics.

## 4.3 Animation of the Simplified Meshes

To validate the animation for rigged characters after simplification, we conduct a series of clips to drive the characters. As shown in Figure 6, the low-poly meshes accurately preserve the integrity of

**Table 1: Evaluation of the meshes by all comparing methods.**

| Method | PSNR↑ | SSIM ↑ | NLPIPS ↑ | IOU3d (%) ↑ |
|---|---|---|---|---|
| | | Tris=1000 | | |
| QEM | 25.39 | 0.84 | 0.81 | 93.08 |
| QEM++ | 28.18 | 0.87 | 0.89 | 95.28 |
| Nvdiff | 29.04 | 0.89 | 0.87 | 95.25 |
| Blender | 27.03 | 0.86 | 0.89 | 94.80 |
| Simplygon | 29.67 | 0.90 | 0.93 | 96.02 |
| **Ours** | **32.34** | **0.93** | **0.95** | **97.31** |
| | | Tris=2000 | | |
| QEM | 26.46 | 0.85 | 0.84 | 95.32 |
| QEM++ | 30.46 | 0.90 | 0.91 | 97.05 |
| Nvdiff | 29.55 | 0.89 | 0.89 | 96.26 |
| Blender | 28.31 | 0.86 | 0.91 | 96.61 |
| Simplygon | 31.74 | 0.92 | 0.94 | 97.19 |
| **Ours** | **33.18** | **0.94** | **0.96** | **98.11** |
| | | Tris=4000 | | |
| QEM | 27.61 | 0.86 | 0.86 | 97.18 |
| QEM++ | 31.89 | 0.93 | 0.94 | 97.64 |
| Nvdiff | 30.13 | 0.89 | 0.90 | 97.10 |
| Blender | 29.77 | 0.90 | 0.92 | 97.45 |
| Simplygon | 33.01 | 0.94 | 0.95 | 98.22 |
| **Ours** | **33.96** | **0.95** | **0.96** | **98.69** |

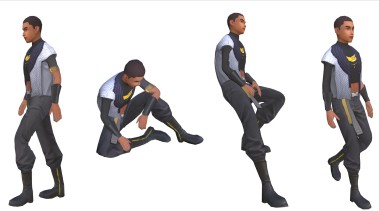

**Figure 6: Animation of the decimated mesh. With Simpli-Guard, even for large movements, the mesh maintains the correctness of the structure.**

the overall shape and exhibit no structural issues even for large deformation. This demonstrates the robustness of SimpliGuard in preserving the essential geometric features for character animation.

## 4.4 Runtime

We run SimpliGuard on a V100 GPU, and the overall time is illustrated in Figure 7. To remesh a 30,000-face mesh to 1,000 faces, SimpliGuard takes 84s totally. In particular, the reconstruction module takes 4.38s, the texture fast initialization module takes 0.33s,

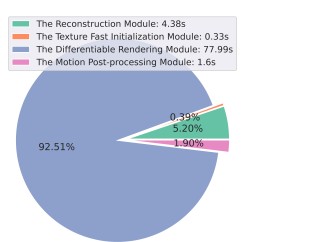
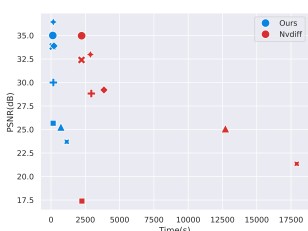
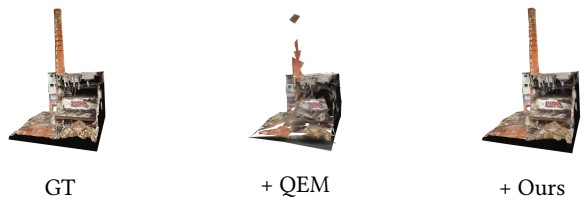

**Figure 7: Runtime. Left: The time consumption of different modules in SimpliGuard (taking the example of reducing from 30,000 faces to 1,000 faces). Right: The time-PSNR graph of SimpliGuard and Nvdiff when reducing to 1,000 faces from different initial numbers of faces.**

GT                    + QEM                    + Ours

**Figure 8: The effectiveness of the reconstruction module. We replace the module with QEM while keeping the other modules unchanged. The mesh structure with the module is significantly superior to the one using QEM, which highlights the necessity of the reconstruction module.**

the differentiable rendering module takes 77.99s, and the motion post-processing module takes 1.60s. For comparison, we also plot the time distribution of Nvdiff in Figure 7. Our method outperforms Nvdiff in terms of both runtime and visual quality.

### 4.5 Ablation Studies

In this section, we validate the necessity and effectiveness of various components in SimpliGuard.

*The Reconstruction Module.* To validate the impact of this module, we replace it with QEM. It takes the reference mesh as input and generates a mesh with the desired number of faces. This mesh then goes through the subsequent modules to obtain the final result. The results, shown in Figure 8, clearly demonstrate that using QEM leads to dissimilarity to the reference mesh and structural issues. In contrast, the structure obtained with the reconstruction module is significantly superior to the QEM-based version both in terms of structure and visual appearance. This emphasizes the necessity of the reconstruction module to achieve desirable results.

*The Texture Fast Initialization Module.* As shown in Figure 9, with the inclusion of this module, SimpliGuard already produces preliminary results at 2s, while without the module, the textures remain significantly blurry. Besides, to achieve the same PSNR, the inclusion of this module takes approximately 1/10 of the time compared to when it is not included. This strongly demonstrates the effectiveness of the module in accelerating convergence.

*The Differentiable Rendering Module.* We conducted ablation experiments on the proposed losses. From Figure 11, it is visually

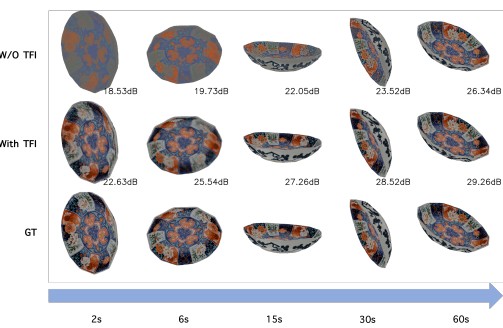

**Figure 9: The effectiveness of the texture fast initialization module. The module significantly improves the convergence speed of the mesh, with an acceleration of 10 times compared to the results without the module.**

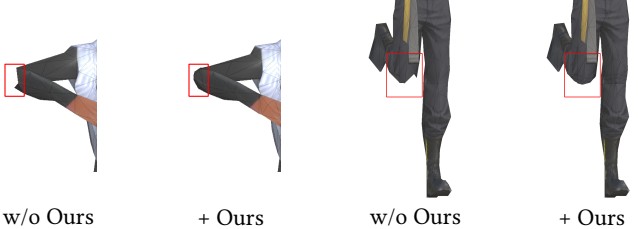

w/o Ours          + Ours          w/o Ours          + Ours

**Figure 10: The effectiveness of the motion post-processing module. Without the module, significant bending can cause structural issues. When this module is included, it can maintain a good structure.**

apparent that the inclusion of the Gradient Loss enhances texture details and reduces the blurriness of texture edges. From Figure 11, with the inclusion of the texture loss, the overall texture transitions become smooth, and no abnormal textures are present.

*The Motion Post-processing Module.* For a low-poly mesh, as shown in Figure 10, incorporating the module ensures the preservation of a robust 3D structure even under animation. In contrast, without the module, some triangles are likely to span across joints, leading to noticeable structural degradation during animation. Therefore, this module plays a crucial role in meshes with skeletons.

## 5 CONCLUSIONS

We have presented our proposed framework, SimpliGuard for handling arbitrary meshes. Compared to previous methods, our approach offers several advantages: 1) No assumption constraints are imposed on the input mesh. 2) It can achieve extreme mesh simplification in a few minutes, resulting in high-quality meshes in terms of both texture and structure. 3) For rigged meshes, SimpliGuard ensures the correctness of the structure during motion even after extreme simplification. These advantages are achieved through the combined efforts of various modules and the designed loss functions. In the experiments, we showcase the significant advantages of our method compared to other approaches by conducting mesh simplification on complex meshes in the wild. This highlights the enormous potential of our method for practical applications.

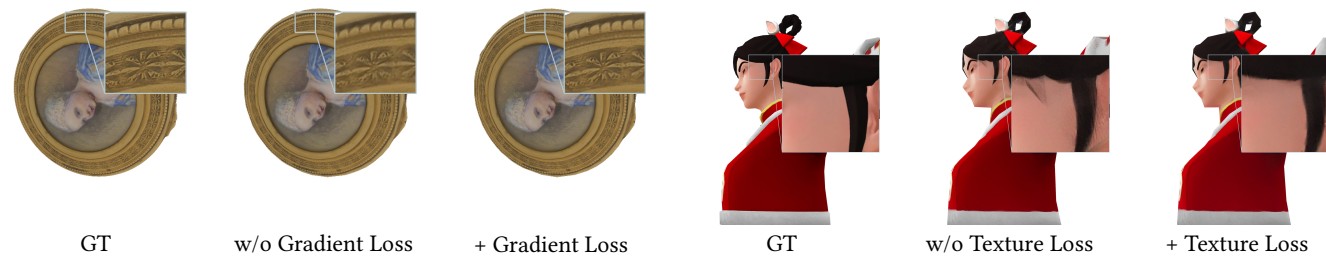

GT          w/o Gradient Loss          + Gradient Loss          GT          w/o Texture Loss          + Texture Loss

**Figure 11: The effectiveness of the differentiable rendering module. Left: Gradient Loss.** $\mathcal{L}_{\mathbf{grad}}$ **significantly enhances the quality of the texture. Right: Texture Loss. Without** $\mathcal{L}_{\mathbf{tex}}$**, the textures appear rough and dirty. By introducing** $\mathcal{L}_{\mathbf{tex}}$**, the generated texture is smoother and better for visual perception.**

GT(999k tris)          QEM(4k tris)          QEM++(238k tris)          Nvdiff(4k tris)          Ours(4k tris)

GT(77k tris)          QEM(1k tris)          QEM++(2.9k tris)          Nvdiff(1k tris)          Ours(1k tris)

**Figure 12: More results of comparison with academic approaches.**

GT(106k tris)          Blender(4k tris)          simplygon(4k tris)          Ours(4k tris)

GT(474k tris)          Blender(4k tris)          simplygon(4k tris)          Ours(4k tris)

**Figure 13: More results of comparison with featured software.**

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
