# OpenReview forum: "SimpliGuard: Robust Mesh Simplification In the Wild"
_acmmm.org/ACMMM/2024/Conference — MM2024 Poster_

### Official Review · Reviewer_4iun · 2024-05-21

**Rating:** 5
**Confidence:** 4

**Summary:**

It proposes a framework, SimpliGuard, for mesh simplification, handling meshes in the wild, and keeping a similar structure, texture, and good moveability after processing.

SimpliGuard consists of four parts: the reconstruction module, the texture initialization module, the differentiable rendering module, and the motion post-processing module. The reconstruction module first simplifies meshes, and the texture initialization module generates coarse texture maps. Then, the differentiable rendering module optimizes the structures and textures of meshes. Finally, the motion post-processing module fine-tunes mesh structures for animation. Comprehensive experiments demonstrate the superiority of SimpliGuard over existing methods.

**Strengths:**

1. It has good motivation and produces meaningful results.
2. It simultaneously solves three key problems in mesh simplification: face reduction, texture keeping, and moveability for animation.

**Limitations:**

1. About meshes in the wild:
   1. It is unclear how and why the proposed method can handle meshes in the wild.
   2. Can the data from Sketchfab represent meshes in the wild?
2. The proposed method cannot handle the non-watertight meshes since the reconstruction module is based on the classical marching cubes.
3. The differentiable rendering takes too much time. Are the benefits of this module worth the time-consuming? I hope to have some discussions or ablation studies about this.
4. There is no discussion about limitations.
5. There are some writing problems:
   - Line 139, no reference for Nvdiff.
   - Lines 261 and 262, M_tar as the input and M_src as the output, are confusing. I suggest using the dense mesh and simplified mesh.
   - Lines 366-370, symbols in the equation overlap.
   - Line 578, there should be a blank between sentences.

**Suitability:**

3

---

### Official Review · Reviewer_oovs · 2024-05-25

**Rating:** 4
**Confidence:** 2

**Summary:**

The paper introduces SimpliGuard, a robust mesh simplification framework for complex meshes. It consists of four modules: reconstruction, texture fast initialization, differentiable rendering, and motion post-processing. The reconstruction module prepares meshes by converting them into manifold structures. The texture fast initialization module quickly generates texture maps to reduce optimization time. The differentiable rendering module enhances structure and texture quality using advanced loss functions. The motion post-processing module ensures the structural integrity of animated meshes. Experiments show SimpliGuard outperforms traditional methods and industry software like Blender and Simplygon. It delivers high-quality results efficiently, suitable for real-world applications. Ablation studies validate the effectiveness of each module.

**Strengths:**

+ SimpliGuard presents a framework for mesh simplification that effectively handles complex, unstructured meshes in the wild. Unlike traditional methods, it combines multiple advanced modules to ensure high-quality results.
+ The framework employs a comprehensive approach by integrating several key components: reconstruction, texture initialization, differentiable rendering, and motion post-processing. Each component addresses specific challenges in mesh simplification.
+ The paper provides a thorough evaluation of SimpliGuard through extensive experiments. It compares the framework with traditional methods (QEM, QEM++) and industry software (Blender, Simplygon), using both qualitative and quantitative metrics (PSNR, SSIM, NLPIPS, 3D IoU).

**Limitations:**

- The framework involves multiple intricate modules and techniques, such as the reconstruction module, texture fast initialization, and differentiable rendering with specialized loss functions. Re-implementing such a framework for real application could be challenging.

- While the paper provides a thorough evaluation against traditional methods (QEM, QEM++) and industry software (Blender, Simplygon), it lacks comparisons with some of the latest neural network-based mesh simplification methods. Including these comparisons could have provided a more comprehensive evaluation of SimpliGuard's performance against the state-of-the-art in machine learning approaches, such as [1].

[1] Chen Z, Pan Z, Wu K, et al. Robust Low-Poly Meshing for General 3D Models[J]. ACM Transactions on Graphics (TOG), 2023, 42(4): 1-20.

- The paper focuses on meshes obtained from Sketchfab, which might not fully represent the diversity of real-world 3D models. Testing SimpliGuard on a broader range of datasets, including those from different sources and with varied characteristics would strengthen the claim of its robustness and generalization capabilities, for example, including more results on Thingi10k dataset [2].

[2] Zhou Q, Jacobson A. Thingi10k: A dataset of 10,000 3d-printing models[J]. arXiv preprint arXiv:1605.04797, 2016.

- Chamfer loss is a commonly used metric to measure the distance between vertices. Is it possible to include Chamfer loss in this method to improve geometric reduction performance?

- The current reported results are not clear enough to adequately demonstrate the quality of the re-meshed results. Can this method avoid issues like inconsistent face orientation and self-intersections in complex topologies?

- It would also be beneficial to include the vertex and face count in each visual result.

- This paper employs the cycle-point strategy to reduce the occurrence of "spots." How about adding a linear interpolation post-processing step to address these spots? It seems that these scattered spots could be easily removed with this simple strategy.

**Suitability:**

2

---

### Official Review · Reviewer_PVqS · 2024-06-03

**Rating:** 4
**Confidence:** 2

**Summary:**

The paper presents a novel framework for mesh simplification which incorporates texture and rigging in to account. The proposed method is structured into four distinct modules, each addressing a specific aspect of the process: reconstruction, texture initialization, differentiable rendering, and motion post-processing. Experiment has been conducted on selected models from Sketchfab with the evaluation based on four metrics: PSNR, SSIM, NLPIPS, IOU3d.

**Strengths:**

+ The paper extends the scope of mesh simplification by incorporating additional aspects such as texture and rigging.
+ The structure of the paper is well-organized and easy to follow.

**Limitations:**

- Given that the objective is mesh simplification, it would be beneficial to include an evaluation of the resulting mesh quality.
- For the evaluation based on rendered results, a detailed examination including zoomed-in views is recommended to adequately assess features such as specular highlights.
- The effectiveness of differentiable rendering module is not straightforward, and the paper does not include an ablation study comparing the results with and without this module.

**Suitability:**

2

---

### Meta-Review · Area_Chair_FiHc · 2024-07-08

**Recommendation:** Accept (Poster)
**Confidence:** 5

**Metareview:**

This paper presents SimpliGuard, a robust mesh simplification framework for complex meshes. It consists of four modules, i.e., reconstruction, texture fast initialization, differentiable rendering, and motion post-processing. All reviewers have positive final ratings, are satisfied with the response, and recommend accepting the paper. I agree with their recommendation. Thanks for the authors' effort and rebuttal.